# Exploring Large Language Models for Multi-Modal Out-of-Distribution Detection

**Yi Dai**[1][*], **Hao Lang**[2][†], **Kaisheng Zeng**[1], **Fei Huang**[2], **Yongbin Li**[2][‡]

[1] Department of Computer Science and Technology, Tsinghua University [2] Alibaba Group

{hao.lang, f.huang, shuide.lyb}@alibaba-inc.com,

{dai-y21, zks19}@mails.tsinghua.edu.cn

## Abstract

Out-of-distribution (OOD) detection is essential for reliable and trustworthy machine learning. Recent multi-modal OOD detection leverages textual information from in-distribution (ID) class names for visual OOD detection, yet it currently neglects the rich contextual information of ID classes. Large language models (LLMs) encode a wealth of world knowledge and can be prompted to generate descriptive features for each class. Indiscriminately using such knowledge causes catastrophic damage to OOD detection due to LLMs' hallucinations, as is observed by our analysis. In this paper, we propose to apply world knowledge to enhance OOD detection performance through selective generation from LLMs. Specifically, we introduce a consistency-based uncertainty calibration method to estimate the confidence score of each generation. We further extract visual objects from each image to fully capitalize on the aforementioned world knowledge. Extensive experiments demonstrate that our method consistently outperforms the state-of-the-art.

## 1 Introduction

Machine learning models deployed in the wild often encounter out-of-distribution (OOD) samples that are not seen in the training phase (Bendale and Boult, 2015; Fei and Liu, 2016). A reliable model should not only obtain high performance on samples from seen distributions, i.e., in-distribution (ID) samples, but also accurately detect OOD samples for caution (Amodei et al., 2016; Boult et al., 2019; Dai et al., 2023b). Most existing OOD detection methods are built upon single-modal inputs, e.g., visual inputs (Hsu et al., 2020; Liu et al., 2020) or textual inputs (Zhou et al., 2021; Zhan et al., 2021). Recently, Esmaeilpour et al. (2022); Ming et al. (2022a) attempt to tackle multi-modal

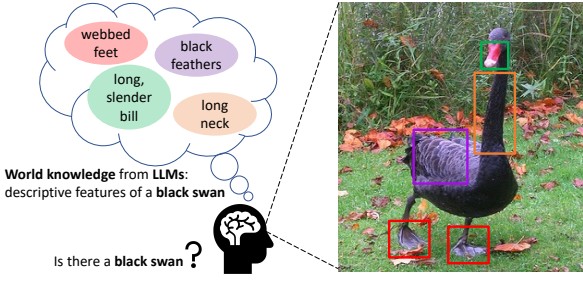

Figure 1: World knowledge from large language models can facilitate the detection of visual objects.

OOD detection problem that explores the semantic information conveyed in class labels for visual OOD detection, relying on large-scale pre-trained vision-language models such as CLIP (Radford et al., 2021).

In this paper, we apply world knowledge from large language models (LLMs) (Petroni et al., 2019a) to multi-modal OOD detection by generating descriptive features for class names (Menon and Vondrick, 2023). As illustrated in Figure 1, to find a *black swan*, look for its *long neck*, *webbed feet*, and *black feathers*. These descriptors provide rich additional semantic information for ID classes, which can lead to a more robust estimation of OOD uncertainty (Ming et al., 2022a), i.e., measuring the distance from the visual features of an input to the closest textual features of ID classes.

However, the knowledge encoding of LLMs such as GPT-3 (Brown et al., 2020) is lossy (Peng et al., 2023) and tends to hallucinate (Ji et al., 2023), which can cause damage when applied for OOD detection tasks. As shown in Figure 2, LLMs generate unfaithful descriptors for class "*hen*", assuming a *featherless head* appearing in a *hen*. Indiscriminately employing generated descriptive features to model ID classes brings noise to the inference process due to LLMs' hallucinations. Moreover, this issue becomes more severe as OOD detection deals with samples in an unbounded feature space (Shen

---

[*] Work done while the author was interning at Alibaba.

[†] Equal contribution.

[‡] Corresponding author.

et al., 2021). Collisions between OOD samples and ID classes with augmented descriptors would be common.

To address the challenge mentioned above, we propose an approach for selective generation of high-quality descriptive features from LLMs, while abstaining from low-quality unfaithful ones (Ren et al., 2022). Recent studies show LLMs can predict the quality of their outputs, i.e., providing calibrated confidence scores for each prediction that accurately reflects the likelihood of the predicted answer being correct (Kadavath et al., 2022; Si et al., 2022a). Unfortunately, descriptors of a class name generated by LLMs are long-form and structured intermediate results for the ultimate OOD detection task, and calibration of LLMs for generating such long open-ended text (Lee et al., 2022) is still in its infancy.

We perform uncertainty calibration in LLMs by exploring a consistency-based approach (Wang et al., 2022). We assume if the same correct prediction is consistent throughout multiple generations, then it could serve as a strong sign that LLMs are confident about the prediction (Si et al., 2022b). Instead of computing literal similarity, we define consistency between multiple outputs from LLMs for a given input based on whether they can retrieve similar items from a fixed set of unlabeled images. Specifically, for each descriptor, we first retrieve a subset of images, leveraging the joint vision-language representations. Then, we measure generation consistency by calculating the overlap between these image subsets.

To further capitalize on the world knowledge expressed in descriptors from LLMs, we employ a general object detector to detect all the candidate objects (concepts) in an image (Cai et al., 2022) and represent them with their predicted class names (Chen et al., 2023b) such as "*mirror*", "*chair*", and "*sink*" (see Figure 5). These visual concepts provide valuable contextual information about an image in the textual space and can potentially match descriptive features of an ID class if the image belongs to that class. Accordingly, we improve our distance metric of input samples from ID classes by considering the similarity between image visual concepts and ID class descriptive features in language representations. Our key contributions are summarized as follows:

- We apply world knowledge from large language models (LLMs) to multi-modal OOD

detection for the first time by generating descriptive features for ID class names.

- We analyse LLMs' hallucinations which can cause damage to OOD detection. A selective generation framework is introduced and an uncertainty calibration method in LLMs is developed to tackle the hallucination issue.

- We detect objects in an image and represent them with their predicted class names to further explore world knowledge from LLMs. Our extensive experimentation on various datasets shows that our method consistently outperforms the state-of-the-art.

## 2 Related Work

**OOD Detection** is widely investigated in vision classification problems (Yang et al., 2021), and also in text classification problems (Lang et al., 2023). Existing approaches try to improve the OOD detection performance by logits-based scores (Hendrycks and Gimpel, 2017a; Liu et al., 2020), distance-based OOD detectors (Lee et al., 2018; Sun et al., 2022), robust representation learning (Winkens et al., 2020; Zhou et al., 2021), and generated pseudo OOD samples (Shu et al., 2021; Lang et al., 2022).

Multi-modal OOD detection is recently studied by Fort et al. (2021a); Esmaeilpour et al. (2022); Ming et al. (2022a), which leverages textual information for visual OOD detection. These works do not explore world knowledge from LLMs.

**Large language models** like GPT3 (Brown et al., 2020) can serve as a knowledge base and help various tasks (Petroni et al., 2019b; Dai et al., 2023a). While some works demonstrate that world knowledge from LLMs can provide substantial aid to vision tasks (Yang et al., 2022) such as vision classification (Menon and Vondrick, 2023), its efficacy in multi-modal OOD detection is currently underexplored. Moreover, as LLMs tend to hallucinate and generate unfaithful facts (Ji et al., 2023), additional effects are needed to explore LLMs effectively.

**Uncertainty Calibration** provides confidence scores for predictions to safely explore LLMs, helping users decide when to trust LLMs outputs. Recent studies examine calibration of LLMs in multiple-choice and generation QA tasks (Kadavath et al., 2022; Si et al., 2022a; Kuhn et al., 2023). In multi-modal OOD detection task, open-ended

| ID dataset | Classification(↑) | | OOD Detection(↓) | |
| --- | --- | --- | --- | --- |
| | CLIP | CLIP+Desp. | CLIP | CLIP+Desp. |
| ImageNet-1k | 64.05 | **68.03** | **42.74** | 48.99 |
| CUB-200 | 56.35 | **57.75** | 7.09 | **4.72** |
| Stanford-Cars | 61.56 | **63.26** | **0.08** | 0.10 |
| Food-101 | 85.61 | **88.50** | **1.86** | 3.86 |
| Oxford-Pet | 81.88 | **86.92** | **1.70** | 3.52 |

Table 1: Effect of class name descriptors from LLMs in classification and OOD detection tasks. CLIP and CLIP+Desp. are VLMs based methods without and with descriptors. Classification is evaluated by accuracy and OOD detection is evaluated by FPR95 (averaged on iNaturalist, SUN, Places, Texture OOD datasets).

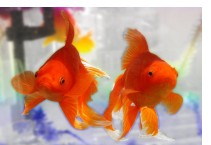
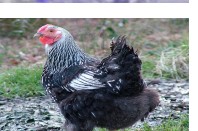
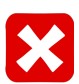

**goldfish**
· long, flowing tail
· small, black eyes
· a small mouth
· dorsal fins

**hen**
· featherless head
· round body shape
· yellow beak
· webbed red-orange legs

Figure 2: Cases of descriptors generated by LLMs. Unfaithful descriptors may appear due to hallucinations.

text (Lee et al., 2022) are generated to provide descriptive features for ID classes (Menon and Vondrick, 2023), and calibration in this task is yet underexplored.

## 3 Background

### 3.1 Problem Setup

We start by formulating the multi-modal OOD detection problem, following Ming et al. (2022a). We denote the input and label space by $\mathcal{X}$ and $\mathcal{Y}$, respectively. $\mathcal{Y}$ is a set of class labels/names referring to the known ID classes. The goal of OOD detection is to detect samples that do not belong to any of the known classes or assign a test sample to one of the known classes. We formulate the OOD detection as a binary classification problem: $G(\mathbf{x}; \mathcal{Y}, \mathcal{I}, \mathcal{T}) : \mathcal{X} \rightarrow \{0, 1\}$, where $\mathbf{x} \in \mathcal{X}$ denotes an input image, $\mathcal{I}$ and $\mathcal{T}$ are image encoder and text encoder from pre-trained vision-language models (VLMs), respectively. The joint vision-language embeddings of VLMs associate objects in visual and textual modalities well. Note that there is no training data of ID samples provided to train the OOD detector.

### 3.2 Analyzing Class Name Descriptors from LLMs

Recent work has demonstrated that class name descriptors, i.e., descriptive features for distinguishing a known object category in a photograph generated by prompting LLMs (see Section 4.2 for more details), can improve zero-shot visual classification performance (Menon and Vondrick, 2023) in a close-world setting (Vapnik, 1991). A natural extension of this work is to leverage the descriptors for OOD detection in an open world (Fei and Liu, 2016), which is largely unexplored.

Unfortunately, we find that the descriptors used in previous approach fail to improve the OOD detection performance in a few datasets. As shown in Table 1, although descriptors can improve the classification performance in all five datasets, they degenerate the OOD detection performance in four ID datasets. We hypothesize this is because LLMs generate unfaithful descriptors due to hallucinations (see cases in Figure 2), which bring noise to the OOD detection process.

To verify our hypothesis, we visualize ID samples from ImageNet-1k dataset and OOD samples from iNaturalist dataset, together with their original class names, based on aligned vision-language features (Radford et al., 2021). As illustrated in Figure 3(b), class names of ID samples may coincide with these of OOD samples, when augmented with descriptors from LLMs. Thus, it is improper to indiscriminately adopt these descriptors.

The above assumptions are also evidenced in Figure 4. ID samples obtain higher similarity scores to their classes when augmented with descriptors (Figure 4(a)), which show the effect of descriptors for the classification task. Meanwhile, OOD samples gain larger maximum similarity scores with ID classes with descriptors (Figure 4(b)), which makes OOD detection more difficult.

## 4 Method

### 4.1 Overview

In this study, we build the multi-modal OOD detector following four steps: **1.** Generate a set of descriptors $\boldsymbol{d}(c)$ for each class name $c \in \mathcal{Y}$ by prompting LLMs; **2.** Estimate a confidence score for descriptors $\boldsymbol{d}(c)$ with uncertainty calibration; **3.** Detect visual objects for each test image $\mathbf{x}$; **4.** Build an OOD detector with selective generation of descriptors and image visual objects. Figure 5 shows an overview of our approach.

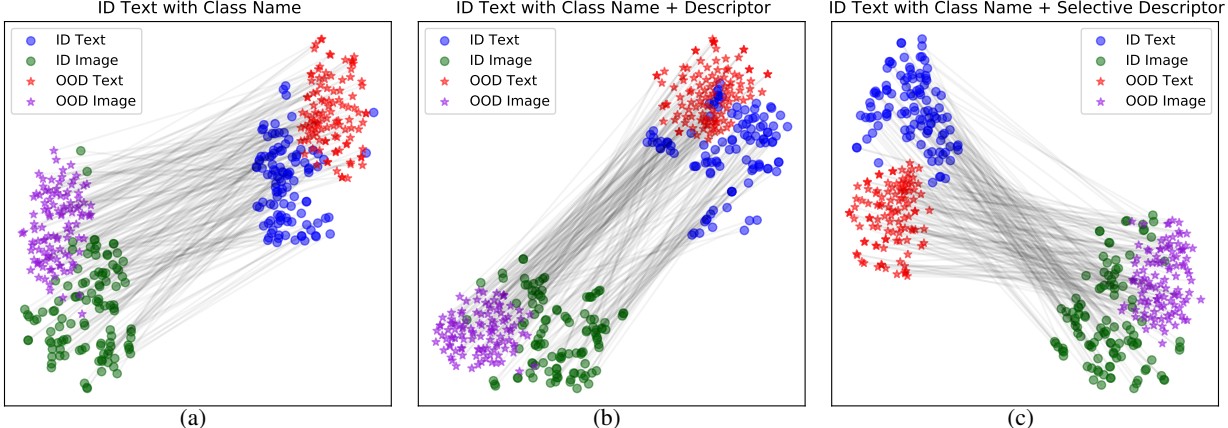

(a)      (b)      (c)

Figure 3: t-SNE visualization of ID and OOD samples, which are from ImageNet-1k and iNaturalist datasets, respectively. ID/OOD Image represent visual representations of images, and ID/OOD Text are textual representations of their class names, based on CLIP. Each line denotes a pair of vision-language representations for one sample.

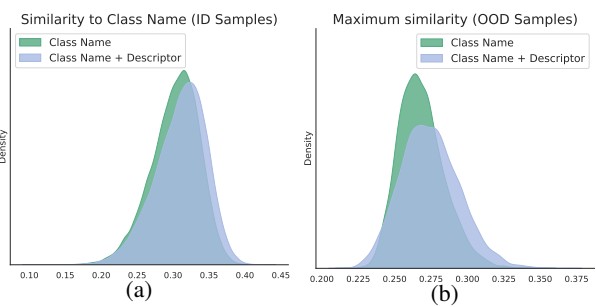

(a)      (b)

Figure 4: Left: Similarities between ID samples and their class names; Right: Maximum similarities between OOD samples and ID classes. ID samples from mageNet-1K and OOD samples from Naturalist.

## 4.2 Descriptor Generation

To apply world knowledge from LLMs for OOD detection, we generate a set of descriptors $d(c)$ for each known class name $c$ by prompting LLMs (see Figure 6), following (Menon and Vondrick, 2023). We randomly select 1 visual category and manually compose descriptors to use as 1-shot in-context example. We prompt LLMs to describe the visual features for distinguishing a category in a photograph. The generated list composes the set $d(c)$. Figure 2 shows cases of generated descriptors, which include shape, size, color, and object parts in natural language.

## 4.3 Uncertainty Calibration

As Figure 2 illustrates, LLMs may generate unfaithful descriptors due to hallucinations, which would hurt the performance of OOD detection if applied indiscriminately. To address this issue, we design a

consistency-based (Wang et al., 2022) uncertainty calibration method to estimate a confidence score for each generation, which helps decide when to trust the LLMs outputs. We assume if the same correct prediction is consistent throughout multiple generations, then it shows that LLMs are confident about the prediction (Si et al., 2022b), thus the generation results are more trustworthy.

It is non-trivial to directly extend previous consistency-based methods to our settings. Specifically, we leverage LLMs to generate long-form and structured descriptor lists without a fixed candidate answer set. Any permutation of the descriptor list can convey the same meaning, despite the inconsistency in surface form. Meanwhile, the outputs of LLMs are intermediate results for our OOD detection task. It is challenging to measure quality of these intermediate results while maintaining a tangible connection to the ultimate detection task.

We take inspiration from prior code generation works (Li et al., 2022; Chen et al., 2023a), which prompt LLMs to generate structured codes aiming at solving programming tasks. Multiple codes are sampled from LLMs for one input and then execution results are obtained by executing these codes. The code with the most frequent execution result is selected as the final prediction. In a similar manner, we quantify the characteristics of descriptor sets through their retrieval feedbacks from a fixed set of unlabeled images, and define their consistency according to consensus among retrieval results. Specifically, we propose a three-stage consistency-based uncertainty calibration method.

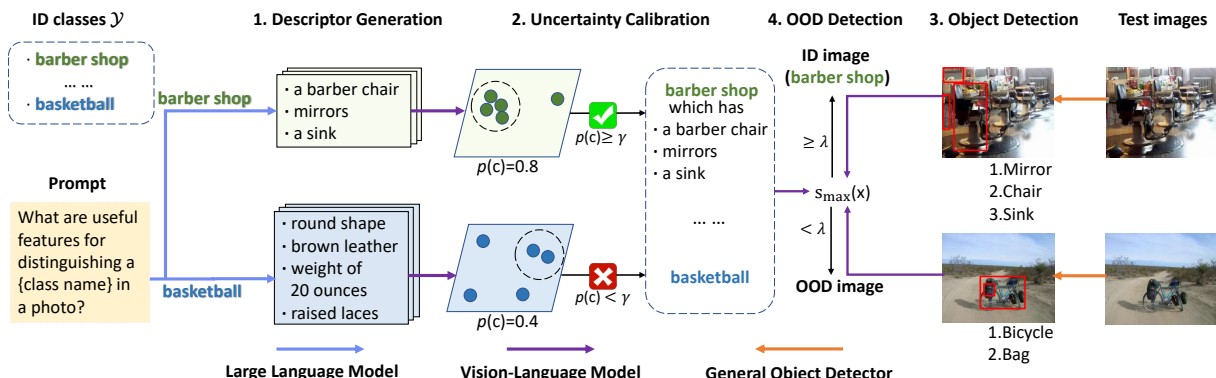

Figure 5: Our multi-modal OOD detection framework. For each image $\mathbf{x}$ and ID classes $\mathcal{Y}$, **1.** Generate a descriptor set for each class $c \in \mathcal{Y}$ by prompting LLMs; **2.** Estimate a confidence score $p(c)$ for each descriptor set; **3.** Detect objects in $\mathbf{x}$ and represent them with object names; **4.** Compute the maximum class matching score $s_{\max}(\mathbf{x})$.

---

| | |
|---|---|
| **Q:** What are useful visual features for distinguishing a *lemur* in a photo? | $Q_{prompt}$ |
| **A:** There are several useful visual features to tell there is a *lemur* in a photo: | |
| - furry bodies | $A_{prompt}$ |
| - long tail | |
| - large eyes | |
| **Q:** What are useful visual features for distinguishing a *goldfish* in a photo? | $Q_{test}$ |
| **A:** There are several useful visual features to tell there is a *goldfish* in a photo: | |
| - long, flowing tail | $A_{test}$ |
| - small, black eyes | |
| - a small mouth | |
| - dorsal fins | |

Figure 6: Example prompt for generating descriptors of the category *goldfish*. The trailing '-' guides LLMs to generate text in the form of a bulleted list.

**Stage I.** We sample $n$ sets of descriptors $\mathcal{D}(c) = \{\boldsymbol{d}_1(c), \cdots, \boldsymbol{d}_n(c)\}$ from LLMs for each ID class name $c \in \mathcal{Y}$.

**Stage II.** We cluster descriptor sets $\mathcal{D}(c)$ into groups $\mathcal{S}(c)$, where each group $\boldsymbol{s} \in \mathcal{S}(c)$ is comprised of descriptor sets that consist with each other. We define descriptor consistency, $C(\cdot, \cdot)$, which retains any two descriptor sets that share the same characteristics through retrieval feedback. Concretely, we retrieve top $k$ images from an unlabeled image set $\mathcal{M}$ via a retriever $R(\cdot)$ for each set $\boldsymbol{d} \in \mathcal{D}(c)$. The resulting image subset for descriptor set $\boldsymbol{d}$ is denoted as $R(\boldsymbol{d}) \in \{0, 1\}^m$, where $m$ is the size of $\mathcal{M}$ and entry $j$ of the vector is 1 if the $j$-th image of $\mathcal{M}$ is in the retrieved subset. Finally, we assume descriptor consistency $C(\boldsymbol{d}, \boldsymbol{d}')$ holds if cosine similarity between $R(\boldsymbol{d})$ and $R(\boldsymbol{d}')$ is above

$\eta$. Note that text similarity between descriptor sets can also be used in consistency computation.

**Stage III.** We compute the confidence score $p(c)$ for descriptor set $\boldsymbol{d}(c)$ as $\frac{|\boldsymbol{s}^*|}{n}$, where $\boldsymbol{s}^*$ is the largest group in $\mathcal{S}(c)$.

### 4.4 Visual Object Detection

To further capitalize on the world knowledge conveyed in generated descriptors, we introduce a general object detector with a vocabulary of 600 object categories to detect visual objects $\boldsymbol{v}(\mathbf{x})$ for each testing image $\mathbf{x}$ (Cai et al., 2022). Specifically, $\boldsymbol{v}(\mathbf{x})$ consists of detected objects' class names, such as "*mirror*", "*chair*", and "*sink*" in a photograph of a barber shop (see Figure 5).

### 4.5 OOD Detection

For each ID class name $c$, descriptor set $\boldsymbol{d}(c)$ is used to augment the representation of $c$ if its confidence score $p(c)$ is above threshold $\gamma$, otherwise $c$ is used to represent that class only. Thus, the textual features for class name $c$ are:

$$\boldsymbol{t}(c) = \begin{cases} \{g(d) | d \in \boldsymbol{d}(c)\}, & \text{if } p(c) \geq \gamma, \\ \{c\}, & \text{otherwise,} \end{cases}$$

where $d$ is one descriptor in the set $\boldsymbol{d}(c)$ and $g(\cdot)$ transforms $d$ into the form {c} which has {d}.

For an input image $\mathbf{x}$, we calculate the class-wise matching score for each ID class name $c \in \mathcal{Y}$:

$$s_c(\mathbf{x}) = \mathop{\mathbb{E}}_{t \in \boldsymbol{t}(c)} \sigma(\mathcal{I}(\mathbf{x}), \mathcal{T}(t)) + \mathop{\mathbb{E}}_{\substack{v \in \boldsymbol{v}(\mathbf{x}) \\ t \in \boldsymbol{t}(c)}} \sigma(\mathcal{T}(v), \mathcal{T}(t)), \tag{1}$$

where $\sigma(\cdot, \cdot)$ denotes the cosine similarity function, the left term computes the similarity between image visual representations and class name textual representations, and the right term measures

the similarity between detected image objects and class names in the text space.

Lastly, we define the maximum class matching score as: $s_{\max}(\mathbf{x}; \mathcal{Y}, \mathcal{I}, \mathcal{T}) = \max_c \frac{\exp(s_c(\mathbf{x}))}{\sum_{c' \in \mathcal{Y}} \exp(s_{c'}(\mathbf{x}))}$, similar to Ming et al. (2022a). Our OOD detection function can be defined as:

$$G(\mathbf{x}; \mathcal{Y}, \mathcal{I}, \mathcal{T}) = \begin{cases} 1 & s_{\max}(\mathbf{x}; \mathcal{Y}, \mathcal{I}, \mathcal{T}) \geq \lambda \\ 0 & s_{\max}(\mathbf{x}; \mathcal{Y}, \mathcal{I}, \mathcal{T}) < \lambda \end{cases}, \quad (2)$$

where 1 represents ID class and 0 indicates OOD conventionally. $\lambda$ is a chosen threshold.

# 5 Experiments

## 5.1 Datasets and Metrics

**Datasets** Following recent works (Ming et al., 2022a), we use large-scale datasets that are more realistic and complex. We consider the following ID datasets: variants of ImageNet (Deng et al., 2009), CUB-200 (Wah et al., 2011), Stanford-Cars (Krause et al., 2013), Food-101 (Bossard et al., 2014), Oxford-Pet (Parkhi et al., 2012). For OOD datasets, we use iNaturalist (Van Horn et al., 2018), SUN (Xiao et al., 2010), Places (Zhou et al., 2017), and Texture (Cimpoi et al., 2014).

**Metrics** For evaluation, we use these metrics (1) the false positive rate (FPR95) of OOD samples when the true positive rate of ID samples is at 95%, (2) the area under the receiver operating characteristic curve (AUROC).

## 5.2 Implementation Details

In our experiments, we adopt CLIP (Radford et al., 2021) as the pre-trained vision-language model. Specifically, we mainly use *CLIP-B/16* (CLIP-B), which consists of a *ViT-B/16* Transformer as the image encoder and a masked self-attention Transformer (Vaswani et al., 2017) as the text encoder. We also use *CLIP-L/14* (CLIP-L) as a representative of large models. To generate descriptors, we query *text-davinci-003* (Ouyang et al., 2022) with sampling temperature $T = 0.7$ and maximum token length of 100. We construct the unlabeled image set $\mathcal{M}$ through the random selection of $m = 50000$ images from the training set of ImageNet. The retriever $R(\cdot)$ retrieves $k = 50$ images from $\mathcal{M}$. We set the threshold $\eta = 0.9$ and $\gamma = 0.5$. In visual object detection, we employ the object detection model *CBNetV2-Swin-Base* (Cai et al., 2022) as a general object detector with a

vocabulary of 600 objects. See more details in Appendix B.

## 5.3 Baselines

We compared our method with competitive baselines: **1. MOS** (Huang and Li, 2021) divides ID classes into small groups with similar concepts to improve OOD detection; **2. Fort et al.** (Fort et al., 2021b) finetunes a full ViT model pre-trained on the ID dataset; **3. Energy** (Liu et al., 2020) proposes a logit-based score to detect OOD samples; **4. MSP** (Hendrycks and Gimpel, 2017b) employs the maximum classification probability of samples to estimate OOD uncertainty; **5. MCM** (Ming et al., 2022a) estimates OOD uncertainty with the maximum similarity between the embeddings of a sample and ID class names; **6. Menon et al.** (Menon and Vondrick, 2023) prompts LLMs to generate descriptors of each class as cues for image classification. We extend it to OOD detection and use the maximum classification probability as a measure of OOD uncertainty (Hendrycks and Gimpel, 2017b).

## 5.4 Main Results

To evaluate the scalability of our method in real-world scenarios, we compare it with recent OOD detection baselines on the ImageNet-1k dataset (ID) in Table 2. It can be seen that our method outperforms all competitive zero-shot methods. Compared with the best-performing zero-shot baseline MCM, it reduces FPR95 by 5.03%. We can also observe that: **1.** Indiscriminately employing knowledge from LLMs (i.e., Menon et al.) degenerates the OOD detection performance. This indicates the adverse impact of LLMs' hallucinations and underlines the importance of selective generation from LLMs. **2.** Despite being training-free, our method favorably matches or even outperforms some strong task-specific baselines that require training (e.g., MOS). It shows the advantage of incorporating world knowledge from LLMs for OOD detection.

We further evaluate the effectiveness of our method on hard OOD inputs. Specifically, two kinds of hard OOD are considered, i.e., semantically hard OOD (Winkens et al., 2020) and spurious OOD (Ming et al., 2022b). As shown in Table 3, our method exhibits robust OOD detection capability and outperforms all competitive baselines, e.g., improvement of 1.93% in FPR95 compared to the best-performing baseline MCM. We can also observe that zero-shot methods generally obtain higher performance than task-specific baselines.

**Table 2**

| Method | OOD Dataset | | | | | | | | Average | |
|---|---|---|---|---|---|---|---|---|---|---|
| | iNaturalist | | SUN | | Places | | Texture | | | |
| | FPR95↓ | AUROC↑ | FPR95↓ | AUROC↑ | FPR95↓ | AUROC↑ | FPR95↓ | AUROC↑ | FPR95↓ | AUROC↑ |
| *Task-specific (training required)* | | | | | | | | | | |
| MOS (BiT) | 9.28 | 98.15 | 40.63 | 92.01 | 49.54 | 89.06 | 60.43 | 81.23 | 39.97 | 90.11 |
| Fort et al.(ViT-B) | 15.07 | 96.64 | 54.12 | 86.37 | 57.99 | 85.24 | 53.32 | 84.77 | 45.12 | 88.25 |
| Fort et al.(ViT-L) | 15.74 | 96.51 | 52.34 | 87.32 | 55.14 | 86.48 | 51.38 | 85.54 | 43.65 | 88.96 |
| Energy (CLIP-B) | 21.59 | 95.99 | 34.28 | 93.15 | 36.64 | 91.82 | 51.18 | 88.09 | 35.92 | 92.26 |
| Energy (CLIP-L) | 10.62 | 97.52 | 30.46 | 93.83 | 32.25 | 93.01 | 44.35 | 89.64 | 29.42 | 93.50 |
| MSP (CLIP-B) | 40.89 | 88.63 | 65.81 | 81.24 | 67.90 | 80.14 | 64.96 | 78.16 | 59.89 | 82.04 |
| MSP (CLIP-L) | 34.54 | 92.62 | 61.18 | 83.68 | 59.86 | 84.10 | 59.27 | 82.31 | 53.71 | 85.68 |
| *Zero-shot (no training required)* | | | | | | | | | | |
| MCM (CLIP-B) | 30.91 | 94.61 | 37.59 | 92.57 | 44.69 | 89.77 | 57.77 | 86.11 | 42.74 | 90.77 |
| MCM (CLIP-L) | 28.38 | 94.95 | 29.00 | 94.14 | 35.42 | 92.00 | 59.88 | 84.88 | 38.17 | 91.49 |
| Menon et al. (CLIP-B) | 41.23 | 92.09 | 44.12 | 91.72 | 49.74 | 88.91 | 60.89 | 85.07 | 48.99 | 89.45 |
| Menon et al. (CLIP-L) | 33.26 | 93.92 | 30.29 | 94.18 | 37.30 | 91.78 | 59.82 | 84.40 | 40.17 | 91.07 |
| w/o Obj. (CLIP-B) | 23.67 | 95.40 | 37.19 | 92.57 | 43.97 | 89.77 | 56.97 | 86.33 | 40.45 | 91.02 |
| w/o Obj. (CLIP-L) | 28.20 | 95.22 | 27.81 | 94.44 | 33.22 | 91.56 | 56.37 | 86.05 | 36.40 | 91.82 |
| w/o Calib. (CLIP-B) | 48.30 | 90.53 | 41.17 | 92.11 | 45.08 | 89.61 | 56.91 | 87.01 | 47.87 | 89.81 |
| w/o Calib. (CLIP-L) | 40.41 | 92.09 | 29.90 | 94.00 | 35.99 | 91.08 | 52.93 | 87.17 | 39.81 | 91.09 |
| w/o Know. (CLIP-B) | 30.19 | 94.81 | 37.39 | 91.56 | 43.63 | 89.76 | 57.30 | 86.17 | 42.13 | 90.58 |
| w/o Know. (CLIP-L) | 28.66 | 94.88 | 33.25 | 93.70 | 40.00 | 91.31 | 59.09 | 85.41 | 40.26 | 91.33 |
| Ours (CLIP-B) | **22.88** | **95.54** | **34.29** | **92.60** | **41.63** | **89.87** | **52.02** | **87.71** | **37.71** | **91.43** |
| Ours (CLIP-L) | **26.47** | **95.10** | **26.35** | **94.56** | **33.13** | **91.77** | **51.77** | **87.45** | **34.43** | **92.22** |

Table 2: OOD detection performance for ImageNet-1k as ID. The performances of all task-specific baselines come from Ming et al. (2022a).

| Method | ID OOD | ImageNet-10 ImageNet-20 | | ImageNet-20 ImageNet-10 | | Waterbirds Spurious OOD | | Average | |
|---|---|---|---|---|---|---|---|---|---|
| | | FPR95↓ | AUROC↑ | FPR95↓ | AUROC↑ | FPR95↓ | AUROC↑ | FPR95↓ | AUROC↑ |
| *Task-specific (training required)* | | | | | | | | | |
| MOS (BiT) | | 24.60 | 95.30 | 41.80 | 92.63 | 78.21 | 78.60 | 48.20 | 88.84 |
| Fort et al. (ViT-B) | | 8.14 | 98.07 | 11.71 | 98.08 | 4.63 | 98.57 | 8.16 | 98.24 |
| Energy (CLIP-B) | | 15.23 | 96.87 | 15.20 | 96.90 | 41.51 | 89.30 | 23.98 | 94.36 |
| MSP (CLIP-B) | | 9.38 | 98.31 | 12.51 | 97.70 | 39.57 | 90.99 | 20.49 | 95.67 |
| *Zero-shot (no training required)* | | | | | | | | | |
| MCM (CLIP-B) | | 5.00 | 98.31 | 12.91 | 98.09 | 5.87 | 98.36 | 7.93 | 98.25 |
| Menon et al. (CLIP-B) | | 5.80 | 98.65 | 13.09 | 98.08 | 5.57 | 98.45 | 8.15 | 98.39 |
| w/o Obj. (CLIP-B) | | 4.70 | 98.71 | 10.78 | 98.25 | 4.88 | 98.46 | 6.79 | 98.47 |
| w/o Calib. (CLIP-B) | | 6.00 | 98.50 | 11.14 | 98.04 | 5.17 | 98.49 | 7.44 | 98.34 |
| w/o Know. (CLIP-B) | | 5.00 | 98.73 | 11.38 | 98.22 | 4.86 | 98.39 | 7.08 | 98.45 |
| Ours (CLIP-B) | | **4.20** | **98.77** | **9.24** | **98.26** | **4.56** | **98.62** | **6.00** | **98.55** |

Table 3: Performance comparison on hard OOD detection tasks.

This indicates that exposing a model to a training set may suffer from bias and spurious correlations. We also make comparisons on a larger number of ID and OOD datasets in Appendix A.

## 5.5 Ablation Studies

**Model Components** Ablation studies are carried out to validate the effectiveness of each main component in our model. Specifically, the following variants are investigated: **1. w/o Obj.** removes the visual object detection step, i.e., only the left term in Eq. 1 is adopted. **2. w/o Calib.** removes the uncertainty calibration step and indiscriminately uses descriptors from LLMs. **3. w/o Know.** only uses class names to represent each class without

descriptors from LLMs. Results in Table 2 and Table 3 show that our method outperforms all the above variants. Specifically, we can observe that: **1.** Incorporating knowledge from LLMs (see w/o Know.) improves the OOD detection performance by 4.42%. This verifies that world knowledge is important in multi-modal OOD detection. **2.** Both uncertainty calibration (see w/o Calib.) and visual object detection (see w/o Obj.) help to improve the OOD detection performance.

**Uncertainty Calibration** To evaluate our proposed uncertainty calibration method, we perform ablation on alternatives: **1. Confidence** (Si et al., 2022a) leverages language model probabilities of

| Categories | Variants | FPR95↓ | AUROC↑ |
|---|---|---|---|
| Uncertainty Calibration | Confidence | 40.81 | 91.16 |
| | Self-consistency | 41.64 | 90.93 |
| | Self-evaluation | 40.83 | 90.82 |
| Visual Object Detection | Class Sim. | 38.55 | 91.05 |
| | Simple Det. | 39.40 | 90.97 |
| Ours | | **37.71** | **91.43** |

Table 4: Ablation variants of uncertainty calibration and visual object detection. We use the average performance on four OOD datasets with ImageNet-1k as ID.

generated descriptors as the confidence score. **2. Self-consistency** (Wang et al., 2022) makes multiple predictions for one input and makes use of the frequency of the majority prediction as the confidence score. **3. Self-evaluation** (Kadavath et al., 2022) asks LLMs to first propose answers and then estimate the probability that these answers are correct. Results in Table 4 show that our uncertainty calibration method performs better than other variants. This further indicates that dedicated uncertainty calibration approaches should be explored to safely explore generations from LLMs.

**Visual Object Detection**   We evaluate the visual object detection module by implementing the following variants: **1. Class Sim.** uses class name $c$ instead of the descriptive features $t(c)$ in the right term of Eq. 1. **2. Simple Det.** adopts a simple object detection model with a smaller vocabulary of 80 objects (Li et al., 2023). As shown in Table 4, our method outperforms the above variants. Specifically, we can observe that: **1.** Calculating the similarity between detected image concept names and ID class names without descriptors degenerates the OOD detection performance. **2.** Using a general object detection model with a large vocabulary of objects helps to improve the performance.

## 5.6   Further Analysis

**Cases of Retrieval Feedback**   We provide a case study where descriptor sets for the same class are similar/dissimilar in textual form. Figure 7 illustrates that even with low textual similarity and variations in textual form, two descriptor sets can have consistent retrieval feedback if they accurately capture the descriptive features of the same object.

**Analysis of Unlabeled Image Set**   Figure 8 shows the effect of unlabeled image set with varying sizes on the OOD detection performance. We compose image subsets either through random

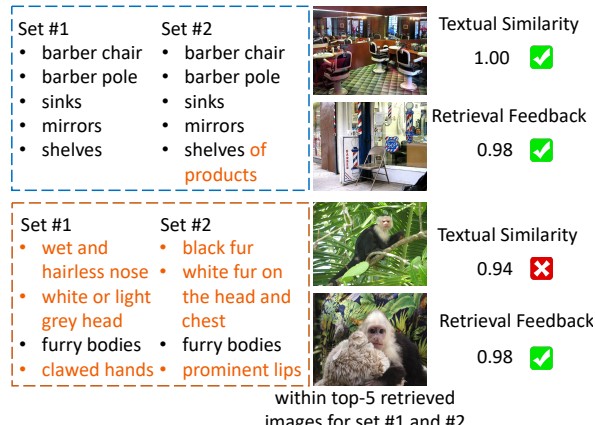

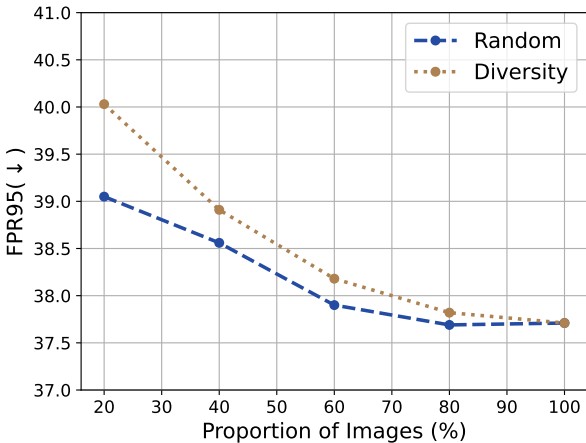

within top-5 retrieved images for set #1 and #2

Figure 7: Case study on descriptor sets and corresponding retrieval feedbacks.

Figure 8: Analysis of unlabeled image set.

down-sampling from the original unlabeled image set ("Random"), or removing images from certain categories ("Diversity"). We can observe that: **1.** Our method achieves superior OOD detection performance along with the increase of unlabeled image data. **2.** Unlabeled image sets that lack diversity achieve limited detection performance, especially in small sizes.

## 6   Conclusion

In this paper, we introduce a novel framework for multi-modal out-of-distribution detection. It employs world knowledge from large language models (LLMs) to characterize ID classes. An uncertainty calibration method is introduced to tackle the issue of LLMs' hallucinations, and visual object detection is proposed to fully capitalize on the generated world knowledge. Experiments on a variety of OOD detection tasks show the effectiveness of our method, demonstrating its exciting property where world knowledge can be reliably exploited

via LLMs by evaluating their uncertainty.

## Limitations

We identify one major limitation of this work is its input modality. Specifically, our method is limited to detecting visual out-of-distribution (OOD) inputs and ignores inputs in other modalities such as textual, audio, electroencephalogram (EEG) and robotic features. These modalities provide valuable information that can be used to construct better OOD detectors. Fortunately, through multi-modal pre-training models (Xu et al., 2021; Huo et al., 2021), we can obtain robust representations in various modalities.

## Ethics Statement

This work does not raise any direct ethical issues. In the proposed work, we seek to develop a zero-shot multi-modal OOD detection model equipped with world knowledge from LLMs, and we believe this work can benefit the field of OOD detection, with the potential to benefit other fields requiring trustworthy models. All experiments are conducted on open datasets.

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

## A More Results

We use an extra collection of ID datasets to showcase the versatility of our method: CUB-200 (Wah et al., 2011), STANFORD-CARS (Krause et al., 2013), FOOD-101 (Bossard et al., 2014), OXFORD-PET (Parkhi et al., 2012), and three variants of ImageNet constructed by Ming et al. (2022a), i.e., ImageNet-10, ImageNet-20, ImageNet-100. The results are shown in Table 5, demonstrating that our method offers superior performance on various multi-modal OOD detection tasks without training.

## B More Implementation Details

In our experiments, we adopt CLIP (Radford et al., 2021) as the pre-trained vision-language model. Specifically, we mainly use *CLIP-B/16* (CLIP-B), which consists of a *ViT-B/16* Transformer as the image encoder and a masked self-attention Transformer (Vaswani et al., 2017) as the text encoder. We also use *CLIP-L/14* (CLIP-L) as a representative of large models. To obtain world knowledge corresponding to each class, we query *text-davinci-003* (Ouyang et al., 2022) with a sampling temperature of 0.7 and a maximum token length of 100.

To obtain retrieval feedback for each descriptor set, We construct a fixed set of unlabeled images, denoted as $\mathcal{M}$, through the random selection of $m = 50000$ images spanning 1000 categories. These images are extracted from the training set of ImageNet-1k without corresponding labels. For descriptor set $\boldsymbol{d}$, the retriever $R(\cdot)$ retrieves top $k$ similar images from $\mathcal{M}$:

$$R'(\boldsymbol{d}) = \operatorname*{argmax}_{M \subset \mathcal{M}, |M| = k} \mathop{\mathbb{E}}_{\substack{\mathbf{x} \in M \\ d \in \boldsymbol{d}}} \sigma(\mathcal{I}(\mathbf{x}), \mathcal{T}(d)).$$

From the retrieved image subset $R'(\boldsymbol{d})$ we derive a binary vector $R(\boldsymbol{d})$, with entry $j$ equal to 1 if the $j$-th image of $\mathcal{M}$ is in $R'(\boldsymbol{d})$. In order to determine whether two descriptor sets, $\boldsymbol{d}$ and $\boldsymbol{d}'$, are consistent with each other, denoted as $C(\boldsymbol{d}, \boldsymbol{d}')$, we incorporate the following two constraints:

$$C(\boldsymbol{d}, \boldsymbol{d}') = \mathbb{1}\Big[\sigma(R(\boldsymbol{d}), R(\boldsymbol{d}')) \geq \eta\Big] \wedge \\ \mathbb{1}\Big[\sigma\big(\frac{\sum_{d \in \boldsymbol{d}} \mathcal{T}(d)}{|\boldsymbol{d}|}, \frac{\sum_{d' \in \boldsymbol{d}'} \mathcal{T}(d')}{|\boldsymbol{d}'|}\big) \geq \eta'\Big], \quad (3)$$

where the first constraint measures the cosine similarity between $R(\boldsymbol{d})$ and $R(\boldsymbol{d}')$, the second constraint computes the cosine similarity between the averaged textual embeddings of descriptors in $\boldsymbol{d}$ and $\boldsymbol{d}'$. Note that the second constraint that computes textual similarity is optional in our method. We evaluate its impact by constructing an ablation variant relying solely on the first constraint. Its average performance on four OOD datasets with ImageNet-1k as ID dataset is 38.59 in FPR95 and 91.37 in AUROC, which outperforms the other zero-shot baselines as well. We set $k = 50$ for image retrieval and $\eta = 0.9$, $\eta' = 0.99$ for consistency computation. We set $\gamma = 0.5$ as the confidence threshold for $p(c)$.

In visual object detection, we use *CBNetV2-Swin-Base* from Cai et al. (2022) with a vocabulary of 600 objects as our general object detector. To construct the ablation variant "Simple Det.", we employ *YOLOv6-L6* from Li et al. (2023) with a smaller vocabulary of 80 categories.

## C Robustness to Sampling Temperature $T$.

We vary sampling temperature $T$ for LLM generation among $\{0.3, 0.5, 0.7, 0.9, 1.1\}$. It can be seen in Figure 9 that regardless of the temperature, our method consistently outperforms the ablation variant "w/o Know." which does not incorporate additional world knowledge from LLMs. We can

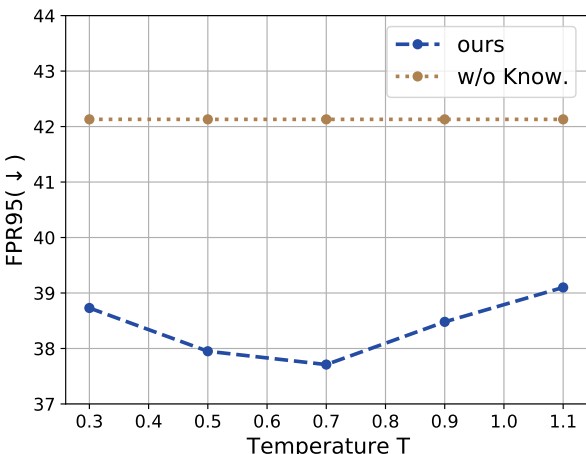

Figure 9: Effect of the sampling temperature $T$ for LLM generation.

also observe that an intermediate temperature of 0.7 can lead to the best performance.

## D  Reliability under Different LLMs, Image Detectors and OOD Detectors

To further verify the reliability of our method, we perform OOD detection using our method under different LLMs (GPT-4, ChatGPT, Claude-1, Claude-2, Bard and text-davinci-003), image detectors (YOLOv6 (Li et al., 2023), InternImage (Wang et al., 2023), Bigdetection (Cai et al., 2022), Co-DETR (Zong et al., 2023)), and OOD detectors (CLIP-based OOD detector without softmax scaling (Ming et al., 2022a)). We use ImageNet-1K as ID dataset, and iNaturalist/SUN/Places/Texture as OOD datasets. As shown in Table 6, our method is reliable when using different LLMs, image detectors and OOD detectors.

| ID Dataset | Method | OOD Dataset | | | | | | | | Average | |
|---|---|---|---|---|---|---|---|---|---|---|---|
| | | iNaturalist | | SUN | | Places | | Texture | | | |
| | | FPR95↓ | AUROC↑ | FPR95↓ | AUROC↑ | FPR95↓ | AUROC↑ | FPR95↓ | AUROC↑ | FPR95↓ | AUROC↑ |
| CUB-200 | MCM | 9.83 | 98.24 | 4.93 | 99.10 | 6.65 | 98.57 | 6.97 | 98.75 | 7.09 | 98.66 |
| | Menon et al. | 8.72 | 98.36 | 3.16 | 99.36 | 3.89 | 99.08 | 3.58 | 99.28 | 4.72 | 99.05 |
| | w/o Obj. | 7.62 | 98.51 | **2.36** | **99.49** | **3.24** | **99.18** | 3.46 | 99.24 | 4.17 | 99.11 |
| | w/o Calib. | 8.85 | 98.09 | 3.60 | 99.29 | 4.64 | 98.92 | 3.09 | 99.38 | 5.04 | 98.92 |
| | w/o Know. | 9.29 | 98.23 | 5.28 | 99.05 | 6.71 | 98.58 | 6.38 | 98.88 | 6.92 | 98.68 |
| | Ours | **2.33** | **99.50** | 3.20 | 99.17 | 7.55 | 98.48 | **2.43** | **99.44** | **3.88** | **99.15** |
| Stanford-Cars | MCM | 0.05 | 99.77 | **0.02** | 99.95 | **0.24** | 99.89 | **0.02** | **99.96** | **0.08** | 99.89 |
| | Menon et al. | 0.07 | **99.82** | 0.05 | **99.96** | 0.28 | **99.90** | **0.02** | **99.96** | 0.10 | **99.91** |
| | w/o Obj. | **0.04** | 99.77 | **0.02** | 99.95 | **0.24** | 99.90 | **0.02** | **99.96** | **0.08** | 99.90 |
| | w/o Calib. | 0.05 | 99.81 | 0.05 | **99.96** | 0.30 | 99.90 | **0.02** | **99.96** | 0.10 | 99.90 |
| | w/o Know. | 0.06 | 99.76 | **0.02** | 99.95 | 0.26 | 99.89 | **0.02** | **99.96** | 0.09 | 99.89 |
| | Ours | 0.05 | 99.75 | **0.02** | **99.96** | **0.24** | 99.90 | **0.02** | **99.96** | **0.08** | **99.91** |
| Food-101 | MCM | 0.72 | 99.76 | 0.90 | **99.75** | **1.86** | 99.58 | 4.04 | 98.62 | 1.86 | 99.43 |
| | Menon et al. | 5.91 | 98.91 | 1.21 | 99.73 | 2.69 | 99.38 | 5.62 | 98.28 | 3.86 | 99.08 |
| | w/o Obj. | 0.72 | 99.77 | 1.02 | 99.74 | 1.93 | 99.55 | 4.17 | 98.63 | 1.96 | 99.42 |
| | w/o Calib. | 6.07 | 98.87 | 1.78 | 99.65 | 3.77 | 99.24 | 5.35 | 98.12 | 4.24 | 98.97 |
| | w/o Know. | 0.76 | 99.76 | 1.06 | 99.73 | 2.12 | 99.54 | 4.20 | 98.59 | 2.04 | 99.40 |
| | Ours | **0.64** | **99.78** | **0.86** | **99.75** | **1.86** | 99.57 | 3.87 | **98.65** | **1.81** | **99.44** |
| Oxford-Pet | MCM | 2.85 | 99.36 | 1.06 | 99.73 | 2.11 | 99.56 | **0.80** | **99.81** | 1.70 | 99.61 |
| | Menon et al. | 8.14 | 98.69 | 1.42 | 99.66 | 3.26 | 99.37 | 1.28 | 99.73 | 3.52 | 99.36 |
| | w/o Obj. | 2.81 | 99.32 | 1.02 | 99.72 | **2.01** | 99.55 | 0.83 | 99.80 | 1.67 | 99.60 |
| | w/o Calib. | 8.41 | 98.61 | 1.45 | 99.65 | 3.27 | 99.35 | 1.31 | 99.72 | 3.61 | 99.33 |
| | w/o Know. | 2.82 | 99.32 | **1.00** | 99.71 | 2.04 | 99.55 | 0.87 | 99.80 | 1.68 | 99.59 |
| | Ours | **2.80** | **99.37** | **1.00** | **99.74** | 2.05 | **99.58** | **0.80** | 99.80 | **1.66** | **99.62** |
| ImageNet-10 | MCM | 0.12 | 99.80 | 0.29 | 99.79 | 0.88 | 99.62 | 0.04 | 99.90 | 0.33 | 99.78 |
| | Menon et al. | 0.13 | 99.81 | 0.28 | 99.82 | 0.90 | 99.66 | 0.04 | 99.91 | 0.34 | 99.80 |
| | w/o Obj. | **0.10** | **99.89** | 0.24 | 99.89 | 0.88 | 99.72 | 0.04 | **99.97** | 0.31 | **99.87** |
| | w/o Calib. | 0.18 | 99.84 | 0.29 | 99.89 | 1.04 | 99.64 | 0.04 | 99.96 | 0.38 | 99.83 |
| | w/o Know. | 0.13 | 99.81 | 0.28 | 99.79 | 0.92 | 99.62 | 0.04 | 99.91 | 0.34 | 99.78 |
| | Ours | 0.12 | 99.87 | **0.22** | **99.90** | **0.83** | **99.73** | **0.02** | **99.97** | **0.30** | **99.87** |
| ImageNet-20 | MCM | 1.02 | 99.66 | 2.55 | 99.50 | 4.40 | 99.11 | 2.43 | **99.03** | 2.60 | 99.32 |
| | Menon et al. | 1.01 | 99.60 | 1.72 | 99.51 | 3.71 | 99.29 | 3.10 | 98.85 | 2.37 | 99.32 |
| | w/o Obj. | 0.97 | 99.59 | 1.72 | 99.51 | 3.61 | 99.29 | 3.05 | 98.87 | 2.35 | 99.32 |
| | w/o Calib. | **0.33** | **99.78** | 2.07 | 99.50 | 3.37 | 99.26 | 2.11 | 98.93 | 1.97 | **99.37** |
| | w/o Know. | 0.75 | 99.71 | 2.08 | 99.52 | 3.67 | 99.20 | 2.43 | 98.83 | 2.23 | 99.32 |
| | Ours | 0.65 | 99.63 | **1.58** | **99.59** | **3.19** | 99.30 | **2.06** | 98.94 | **1.87** | **99.37** |
| ImageNet-100 | MCM | **18.13** | 96.77 | 36.45 | 94.54 | 34.52 | 94.36 | 41.22 | 92.25 | 32.58 | 94.48 |
| | Menon et al. | 22.83 | 96.24 | 27.00 | 95.41 | 31.24 | 94.55 | 42.55 | 92.08 | 30.91 | 94.57 |
| | w/o Obj. | 18.38 | 96.65 | 25.25 | 95.65 | 30.27 | 94.68 | 41.76 | 92.18 | 28.92 | 94.79 |
| | w/o Calib. | 21.73 | 96.38 | 24.35 | 95.75 | 28.07 | 94.93 | 39.06 | 92.70 | 28.30 | 94.94 |
| | w/o Know. | 18.48 | 96.61 | 31.09 | 95.29 | 29.71 | 95.16 | 39.07 | 92.83 | 29.59 | 94.97 |
| | Ours | **18.13** | 96.76 | **22.02** | **96.14** | **26.52** | 95.21 | **38.65** | **93.01** | **26.33** | **95.28** |

Table 5: Zero-shot OOD detection performance based on CLIP-B/16 with various ID datasets.

| | Method | OOD Dataset | | | | | | | | Average | |
|---|---|---|---|---|---|---|---|---|---|---|---|
| | | iNaturalist | | SUN | | Places | | Texture | | | |
| | | FPR95↓ | AUROC↑ | FPR95↓ | AUROC↑ | FPR95↓ | AUROC↑ | FPR95↓ | AUROC↑ | FPR95↓ | AUROC↑ |
| | MCM | 30.91 | 94.61 | 37.59 | 92.57 | 44.69 | 89.77 | 57.77 | 86.11 | 42.74 | 90.77 |
| Different LLMs | Ours(GPT-4) | 22.56 | 95.52 | **33.49** | 92.81 | **40.87** | **90.53** | 51.89 | 87.66 | 37.20 | 91.63 |
| | Ours(ChatGPT) | **21.97** | **95.67** | 33.68 | **92.88** | 41.50 | 90.14 | **50.96** | **88.09** | **37.03** | **91.70** |
| | Ours(Claude-1) | 24.92 | 95.63 | 34.97 | 92.60 | 42.72 | 89.72 | 54.13 | 86.94 | 39.19 | 91.22 |
| | Ours(Claude-2) | 25.83 | 95.08 | 34.71 | 92.65 | 41.65 | 90.86 | 53.44 | 87.02 | 38.91 | 91.40 |
| | Ours(Bard) | 25.74 | 95.07 | 34.17 | 92.72 | 42.12 | 89.85 | 53.48 | 87.02 | 38.88 | 91.17 |
| | Ours(text-davinci-003) | 22.88 | 95.54 | 34.29 | 92.60 | 41.63 | 89.87 | 52.02 | 87.71 | 37.71 | 91.43 |
| Different Image Detectors | Ours(InternImage) | 23.41 | 95.45 | 35.10 | 92.63 | 42.13 | **89.90** | 53.65 | 87.30 | 38.57 | 91.32 |
| | Ours(Co-DETR) | **22.10** | **95.69** | **33.48** | 92.78 | 42.92 | 89.80 | **51.77** | **87.78** | **37.57** | **91.51** |
| | Ours(YOLOv6) | 23.72 | 95.42 | 34.29 | 92.65 | 43.26 | 89.84 | 56.32 | 86.82 | 39.40 | 91.18 |
| | Ours(Bigdetection) | 22.88 | 95.54 | 34.29 | 92.60 | **41.63** | 89.87 | 52.02 | 87.71 | 37.71 | 91.43 |
| Different OOD Detectors | CLIP-based(w/o softmax) | 61.66 | 89.31 | 64.39 | 87.43 | 63.67 | 85.95 | 86.61 | 71.68 | 69.08 | 83.59 |
| | Ours(w/o softmax) | **59.87** | **89.65** | **61.79** | **87.90** | **60.20** | **86.27** | **78.67** | **72.84** | **65.13** | **84.17** |

Table 6: Zero-shot OOD detection performance using different LLMs, image detectors and OOD detectors.