# OpenReview forum: "Exploring Large Language Models for Multi-Modal Out-of-Distribution Detection"
_EMNLP/2023/Conference — EMNLP 2023 Findings_

### Official Review · Reviewer_qF8f · 2023-07-29

**Soundness:** 2

**Excitement:**

3: Ambivalent: It has merits (e.g., it reports state-of-the-art results, the idea is nice), but there are key weaknesses (e.g., it describes incremental work), and it can significantly benefit from another round of revision. However, I won't object to accepting it if my co-reviewers champion it.

**Paper Topic And Main Contributions:**

This paper is a study on a multimodal out-of-distribution detection task, where it addresses the problem using a combination of text and image modalities with the help of a vision language model. The paper proposes a method to improve OOD detection performance by extracting high-quality descriptors related to each class label and utilizing them as additional information source to capture anomalous que. Specifically, it extracts multiple descriptors for each class label and calibrates them in multiple steps to obtain higher quality and faithful descriptors. Through experiments, the proposed method demonstrates better performance compared to existing approaches and shows that each component of the model contributes to performance improvement.

**Questions For The Authors:**

- The validation of the hypotheses presented by the authors is not sufficiently concrete. In lines 207-210, the authors hypothesize that LLM generates unfaithful descriptors, which may negatively impact OOD detection. To verify this hypothesis they conduct experiments in lines 211-227. However, we can verify the difference when descriptors are used or not . In other words, those experiments do not directly confirm the correlation with faithfulness. Consequently, the relationship between descriptor use and OOD performance can be confirmed, but it is difficult to establish a clear connection with unfaithfulness as a variable. There could be many factors at play besides faithfulness.  For example, when the generated descriptors are too general, we can also explain the steep increase in the OOD input.

**Reasons To Accept:**

- The methodology demonstrates good performance without additional training.
- Particularly the method for calibrating long-form and structured descriptors, which is both novel and sensible.
- Various methodologies were diligently compared through experiments, and the strengths of each component are well illustrated from ablations.

**Reasons To Reject:**

- The explanation of specific experimental setups is somewhat lacking, often showing tables or figures without a clear description. This makes it difficult to evaluate the validity of the experiments conducted by the authors:
1. Specifically, in lines 203-206, the description of the experimental setup is missing, and only performance results are presented. There is no explanation of how the descriptors were extracted, or how they were utilized for classification or OOD detection.
2. Similarly, in lines 211-217, Fig3, there is no explanation of how the representations for each input were obtained, or how the descriptors were used in this process.
3. In lines 220-227 (Fig4), it is unclear how the similarity was measured. According to Fig2, there are various descriptors for a single object, but it is not evident how a single similarity score was derived from these diverse descriptors.

- Utilizing a third-party model to create descriptors seems to dilute the advantages of the model.

**Reproducibility:**

4: Could mostly reproduce the results, but there may be some variation because of sample variance or minor variations in their interpretation of the protocol or method.

**Reviewer Confidence:**

4: Quite sure. I tried to check the important points carefully. It's unlikely, though conceivable, that I missed something that should affect my ratings.

**Typos Grammar Style And Presentation Improvements:**

- The paper is difficult to read due to frequent back-and-forth references to figures and tables in the early parts (pg 1-3).

---

> ### Author Rebuttal · Authors · 2023-08-28
>
> Thank you for your detailed, helpful feedback! We’re glad that you found it "good performance without additional training", "novel and sensible", and "diligently compared through experiments". We address your thoughts point by point below.
>
> --"The explanation of specific experimental setups is somewhat lacking, often showing tables or figures without a clear description."
>    * "in lines 203-206, the description of the experimental setup is missing, and only performance results are presented. There is no explanation of how the descriptors were extracted, or how they were utilized for classification or OOD detection": Kindly note that we describe the experimental setup in the caption of Table 1, including ID and OOD datasets, CLIP based feature extractor, and the metrics. We describe the experimental results and conclusions of Table 1 in lines 203-206 without repeating the setup due to space limit. We detail the descriptor generation process in Section 4.2, and we reference this in lines 194-195.
>    * " in lines 211-217, Fig3, there is no explanation of how the representations for each input were obtained, or how the descriptors were used in this process": We describe the representation extraction process in the caption of Figure 3, including ID and OOD datasets, CLIP based feature extractor, and t-SNE visualization. We reference Figure 3 in lines 211-217 and describe what we can learn from it.
>    * "In lines 220-227 (Fig4), it is unclear how the similarity was measured.": We apologize for any confusions. We describe how we measured the similarity between samples and class names augmented with descriptors in Eq. 1 (Section 4.5). Your feedback matters to us and we will provide elaborated details in our revised paper.
>
>
> --"Utilizing a third-party model to create descriptors seems to dilute the advantages of the model."
>
> 1. We apply world knowledge from LLMs to multi-modal OOD detection for the first time.
> 2. We analyse LLMs’ hallucinations which can cause damage to OOD detection.
> 3. We develop a consistency-based uncertainty calibration method to selectively generate high-quality descriptors from LLMs to tackle the above issue.
>
> --"The validation of the hypotheses presented by the authors is not sufficiently concrete. In lines 207-210, the authors hypothesize that LLM generates unfaithful descriptors, which may negatively impact OOD detection. To verify this hypothesis they conduct experiments in lines 211-227. However, we can verify the difference when descriptors are used or not . In other words, those experiments do not directly confirm the correlation with faithfulness. Consequently, the relationship between descriptor use and OOD performance can be confirmed, but it is difficult to establish a clear connection with unfaithfulness as a variable. There could be many factors at play besides faithfulness. For example, when the generated descriptors are too general, we can also explain the steep increase in the OOD input."
>
> 1. We'd like to clarify that faithfulness reflects whether LLM adheres to the source input (Ji et al., ACM Computing Surveys 2023). We prompted the LLMs to generate "useful visual features for distinguishing a category in a photo". Hence, faithful descriptors are required to be precise and distinguishable for each class name. Analysis in lines 211-227 demonstrates that:
>    * First, descriptors provide useful features, enhancing the similarity between ID classes and their samples (Figure 4a).
>    * Second, some descriptors are not distinguishable and bring in noise features (Figure 3,4b), leading to collisions between ID and OOD samples.
>
> 2. We will evaluate the faithfulness of descriptors by manual annotation, and report the results and their connections with detection performence degeneration in the revised paper.

---

### Official Review · Reviewer_ReNH · 2023-08-04

**Soundness:** 4

**Excitement:**

4: Strong: This paper deepens the understanding of some phenomenon or lowers the barriers to an existing research direction.

**Paper Topic And Main Contributions:**

This paper focuses on improving out-of-distribution (OOD) detection in multi-modal machine learning models by leveraging world knowledge from large language models (LLMs). The authors propose generating descriptive features for in-distribution (ID) class names using LLMs, which provide rich semantic information that can enhance OOD detection performance. However, LLMs can hallucinate and generate unfaithful facts, which can negatively impact OOD detection. To address this issue, the authors introduce a consistency-based uncertainty calibration method to estimate the confidence score of each generation. Additionally, they extract visual objects from images to capitalize on the world knowledge expressed in the generated descriptors.

Contribution are:
1. It applies world knowledge from LLMs to multi-modal OOD detection for the first time by generating descriptive features for ID class names.
2. The authors analyze LLMs' hallucinations, which can cause damage to OOD detection, and propose a selective generation framework and an uncertainty calibration method in LLMs to tackle this issue.
3. The paper detects objects in an image and represents them with their predicted class names to further explore world knowledge from LLMs
4. Extensive experimentation on various datasets demonstrates that the proposed method consistently outperforms the state-of-the-art.

**Questions For The Authors:**

1. How to select an object detector that has the right label list to achieve good performance in a new domain (not necessarily covered by common COCO 80 classes)

2. what is the inference speed of the entire system?

3. Visual cues of an object type might be ambiguous (there are numerous types of "birds"). Using LLM to generate key visual look may lead to ambiguous features?

**Reasons To Accept:**

1. The paper addresses a novel and important problem in multi-modal OOD detection by leveraging world knowledge from LLMs
2. The proposed uncertainty calibration method effectively tackles the hallucination issue in LLMs, which is crucial for improving OOD detection performance
3. The use of visual objects extracted from images provides valuable contextual information and enhances the overall approach
4. The experimental results show consistent improvements over the state-of-the-art, demonstrating the effectiveness of the proposed method

**Reasons To Reject:**

1. using LLM to detect OOD means the LLM has strong knowledge of the given class name. This may be limiting to common object types that can be understood by LLM.

**Reproducibility:**

4: Could mostly reproduce the results, but there may be some variation because of sample variance or minor variations in their interpretation of the protocol or method.

**Reviewer Confidence:**

3: Pretty sure, but there's a chance I missed something. Although I have a good feel for this area in general, I did not carefully check the paper's details, e.g., the math, experimental design, or novelty.

---

> ### Author Rebuttal · Authors · 2023-08-28
>
> Thank you for your attentive comments! We are glad you thought "the paper addresses a novel and important problem", and noted the effectiveness of our method. We address your feedback point by point below.
>
> --"using LLM to detect OOD means the LLM has strong knowledge of the given class name. This may be limiting to common object types that can be understood by LLM."
>
> 1. LLMs contain broad world knowledge of massive object types, which have been shown to enhance vision classification performance consistently and significantly (Menon et al.,ICLR2023).
>
> 2. We also evaluate our OOD detection method on mutiple realistic and complex datasets with different distributions, and the experimental results are shown in Table 2 (large-scale datasets), Table 3 (semantically and spurious hard OOD datasets), and Table 5 (larger number of ID and OOD datasets). All these experiments demonstrate that our method consistently outperforms the state-of-the-art.
>
> 3. Meanwhile, we propose a consistency-based uncertainty calibration method in case LLM does not understand the object type, which is a key contribution of our work.
>
> --"How to select an object detector that has the right label list to achieve good performance in a new domain (not necessarily covered by common COCO 80 classes)"
>
> 1. We use a general object detector with a large vocabulary to detect image objects aiming at fully capitalizing on the world knowledge from LLMs, rather than ensuring all the objects in an image are detected.
> 2. We evaluate the reliability of our method with different detectors (YOLOv6 trained with COCO, 80 classes; InternImage trained with OpenImage, 600 classes; Bigdetection, 600 classes; Co-DETR trained with LVIS, 1203 classes). Metrics are FPR95/AUROC. It can be seen that using a general object detection model helps to improve the performance consistently. Even though most objects in samples from Texture are not covered by COCO (only 26 classes are detected by YOLOv6), YOLOv6 still improves the FPR95 score in Texture by 0.65.
>
> |Method|iNaturalist|SUN|Places|Texture|Average|
> |--|--|--|--|--|--|
> |MCM| 30.91/94.61       | 37.59/92.57     | 44.69/89.77     | 57.77/86.11     | 42.74/90.77     |
> |Ours(InternImage)|23.41/95.45|35.10/92.63|42.13/**89.90**|53.65/87.30|38.57/91.32|
> |Ours(Co-DETR)|**22.10**/**95.69**|**33.48**/**92.78**|42.92/89.80|**51.77**/**87.78**|**37.57**/**91.51**|
> |Ours(YOLOv6)|23.72/95.42|34.29/92.65|43.26/89.84|56.32/86.82|39.40/91.18|
> |Ours(Bigdetection)|22.88/95.54|34.29/92.60|**41.63**/89.87|52.02/87.71|37.71/91.43|
>
> 3. We will further explore object detectors with open vocabulary, which enables the model to detect unseen object categories and provides more contextual information.
>
>
> --"what is the inference speed of the entire system?"
> 1. We measured the inference speed with ImageNet-1K as ID dataset, and iNaturalist,SUN,Places,Texture as OOD datasets, which have 85640 samples totally. Our method takes 1194.83s(14ms/sample), and the baseline MCM takes 783.02s(9ms/sample). We will include these details in our revised paper.
>
> --"Visual cues of an object type might be ambiguous (there are numerous types of "birds"). Using LLM to generate key visual look may lead to ambiguous features?"
> 1. You got that right! LLMs are required to generate visual features for **distinguishing** an object category. We also found LLMs may generate unfaithful features due to hallucinations (Figure 2), which can result in collisions between OOD samples and ID classes. In this light, we apply self-consistency to estimate the confidence of LLMs and selectively generate high-quality descriptive features.

---

### Official Review · Reviewer_QiKW · 2023-08-07

**Typos Grammar Style And Presentation Improvements:** No
**Soundness:** 3

**Excitement:**

4: Strong: This paper deepens the understanding of some phenomenon or lowers the barriers to an existing research direction.

**Missing References:**

No

**Paper Topic And Main Contributions:**

Paper summary

Out-of-distribution is an important topic in reliable AI. Multi-modal OOD, which leverages both text and vision information to detect OOD, has not been fully studied yet. In this paper, the authors adopt the Large Language Model on multi-modal OOD by generating descriptive features for class names. To mitigate the hallucination issue, the authors also adopt a consistency-based confidence estimation method to select high-quality descriptions. A general object detector detects all the candidate objects in the image and detects the OOD by comparing the detected object and generated description from LLM.

To evaluate the effectiveness of the proposed method, the authors conduct the experiment on several datasets and comparing with 6 baseline methods.

Presentation: Good
This paper is well-written and easy to follow.

Novelty: Fair
This paper is the first work to adopt LLM to generate descriptions for multi-modal OOD detection.

Soundness: Poor
The biggest issue is the reliability of  LLM and image detector, which may affect the soundness of the method.

Significant: Fair
The proposed method can outperform but can not significantly and consistently outperform other methods.

**Questions For The Authors:**

1. Your method highly relies on the LLM and image detector. Although you have designed a consistency-based confidence estimation method, it is well-known that LLM still suffers from hallucination issues, which will significantly affect the soundness of your method.
2. Can you split your Table 2 into multiple tables or subtables, e.g. one for comparing on CLIP-B, one for comparing on CLIP-L, and one for ablation study?
3. You have adopted the text-davinci-003 to generate the description. Why don't you try the ChatGPT and GPT-4?

**Reasons To Accept:**

1. This paper is well-written and easy to follow.
2. The method that adopts the knowledge from LLM for downstream tasks is a promising direction.
3. The evaluation is comprehensive.

**Reasons To Reject:**

1. The reliability of LLM and image detector is an issue for the soundness of this paper.
2. Table 2 has too much information to show the significance of this paper. It will be much better if the ablation study can be shown in a new table.

**Reproducibility:**

3: Could reproduce the results with some difficulty. The settings of parameters are underspecified or subjectively determined; the training/evaluation data are not widely available.

**Reviewer Confidence:**

4: Quite sure. I tried to check the important points carefully. It's unlikely, though conceivable, that I missed something that should affect my ratings.

---

> ### Author Rebuttal · Authors · 2023-08-28
>
> Thank you for the positive feedback and useful suggestions! We are glad you find it to be "well-written", "the first work to adopt LLM for multi-modal OOD detection," and noted the empirical benefits. We address your thoughts point by point below.
>
> --"The reliability of LLM and image detector is an issue"
>
> 1. We thought the reliability is an important point to evaluate our proposed method. To measure the reliability, we conduct experiments on multiple datasets with different distributions, and the experimental results are shown in Table 2 (large-scale datasets), Table 3 (semantically and spurious hard OOD datasets), and Table 5 (larger number of ID and OOD datasets). All these experiments demonstrate that our method consistently outperforms the state-of-the-art.
>
> 2. To further verify the reliability of our method, we perform OOD detection using our method under different LLMs, image detectors, and OOD detectors.We use ImageNet-1K as ID dataset, and iNaturalist/SUN/Places/Texture as OOD datasets. (Metrics are FPR95/AUROC)
>     * Our method is reliable under diferent LLMs: We generate descriptions from various LLMs (GPT-4, ChatGPT, Claude-1, Claude-2, Bard and text-davinci-003), and our method consistently outperforms the strongest baseline MCM (Ming et al., 2022).
>     * Our method is reliable under diferent image detectors: We detect objects from images using various image detectors (YOLOv6 [1], InternImage [2], Bigdetection, Co-DETR [3]), and our method still consistently outperforms MCM.
>     * Our method is reliable under different OOD detectors: We also apply our method on CLIP-based OOD detector without softmax scaling (Ming et al., 2022), which can enhance its OOD detection performance using descriptors from LLMs.
>
> [1].Li, C., Li, L., Jiang, H., Weng, K., Geng, Y., Li, L., ... & Wei, X. (2022). YOLOv6: A single-stage object detection framework for industrial applications. arXiv preprint arXiv:2209.02976.
>
> [2].Wang, W., Dai, J., Chen, Z., Huang, Z., Li, Z., Zhu, X., ... & Qiao, Y. (2023). Internimage: Exploring large-scale vision foundation models with deformable convolutions. In Proceedings of the IEEE/CVF Conference on Computer Vision and Pattern Recognition (pp. 14408-14419).
>
> [3].Zong, Z., Song, G., & Liu, Y. (2022). Detrs with collaborative hybrid assignments training. arXiv preprint arXiv:2211.12860.
>
> |Group| Method | iNaturalist | SUN       | Places    | Texture   | Average   |
> |---| ---------------------------- | ----------- | --------- | --------- | --------- | --------- |
> || MCM                          | 30.91/94.61       | 37.59/92.57     | 44.69/89.77     | 57.77/86.11     | 42.74/90.77     |
> |**Different LLMs**| Ours(GPT-4)                  | 22.56/95.52       | **33.49**/92.81 | **40.87**/**90.53** | 51.89/87.66     | 37.20/91.63     |
> || Ours(ChatGPT)                | **21.97**/**95.67**   | 33.68/**92.88**     | 41.50/90.14     | **50.96**/**88.09** | **37.03**/**91.70** |
> || Ours(Claude-1)                  | 24.92/95.63       | 34.97/92.60 | 42.72/89.72 | 54.13/86.94     | 39.19/91.22     |
> || Ours(Claude-2)                  | 25.83/95.08       | 34.71/92.65 | 41.65/90.86 | 53.44/87.02     | 38.91/91.40     |
> || Ours(Bard)                  | 25.74/95.07       | 34.17/92.72 | 42.12/89.85 | 53.48/87.02     | 38.88/91.17     |
> || Ours(text-davinci-003)|22.88/95.54|34.29/92.60|41.63/89.87|52.02/87.71|37.71/91.43|
> |**Different image detectors**|Ours(InternImage)|23.41/95.45|35.10/92.63|42.13/**89.90**|53.65/87.30|38.57/91.32|
> ||Ours(Co-DETR)|**22.10**/**95.69**|**33.48**/**92.78**|42.92/89.80|**51.77**/**87.78**|**37.57**/**91.51**|
> ||Ours(YOLOv6)|23.72/95.42|34.29/92.65|43.26/89.84|56.32/86.82|39.40/91.18|
> ||Ours(Bigdetection)|22.88/95.54|34.29/92.60|**41.63**/89.87|52.02/87.71|37.71/91.43|
> |**Different OOD detectors**|CLIP-based(w/o softmax)|61.66/89.31|64.39/87.43|63.67/85.95|86.61/71.68|69.08/83.59|
> ||Ours(w/o softmax)|**59.87**/**89.65**|**61.79**/**87.90**|**60.20**/**86.27**|**78.67**/**72.84**|**65.13**/**84.17**|
>
> --"Table 2 has too much information to show the significance of this paper."
>
> 1. Thank you for this constructive suggestion. We will show main results and ablation studies separately in our revised paper.
>
> --"Although you have designed a consistency-based confidence estimation method, it is well-known that LLM still suffers from hallucination issues, which will significantly affect the soundness of your method."
>
> 1. You make a good point that our method would be affected by hallucination issues of LLMs, which is also evidenced in Section 3.2 of our paper. Hence, we propose a consistency-based uncertainty calibration method for selective generation of descriptors (Section 4).
>
> 2. To evaluate the effectiveness and reliability of our method, we perform experiments under different distributional datasets, LLMs, image detectors, and OOD detectors (See the first responce). All these results show that our method can outperform the state-of-the-art.
>
> --"Can you split your Table 2 into multiple tables or subtables, e.g. one for comparing on CLIP-B, one for comparing on CLIP-L, and one for ablation study?"
>
> 1. Thank you for your constructive feedback. We will refine this for better presentation.
>
> --"You have adopted the text-davinci-003 to generate the description. Why don't you try the ChatGPT and GPT-4?"
>
> 1. Thank you for your valuable suggestion. We evaluate the performance of our method with text-davinci-003, ChatGPT, GPT-4, Claude-1, Claude-2 and Bard to verify the reliability (See the first responce).
>
> --"The proposed method can outperform but can not significantly and consistently outperform other methods."
>
> 1. We compare our method with the strongest zero-shot baseline MCM on multiple datasets. It consistently outperforms MCM by a large margin, e.g., it reduces FPR95 by 5.03% in Table 2.
> 2. Outperforming all task-specific methods (training required) does pose a challenge. Nonetheless, our method outperforms all task-specific baselines except Energy (37.71 vs. 35.92 in FPR95) in Table 2. Our method outperforms all task-specific baselines in Table 3.
> 3. We'd like to kindly point out that reviewer ReNH also endorses our method "consistently outperforms the state-of-the-art."
>
> Given these encouraging new results, we hope you might be convinced to increase your score. We appreciate your feedback strengthening our paper!

---

### Meta-Review · Area_Chair_PdmU · 2023-09-19

**Recommendation:** 2

**Metareview:**

This paper is a study on a multimodal out-of-distribution detection task, where it addresses the problem using a combination of text and image modalities with the help of a vision language model. With the help of large language models, we are able to imagine some of the image descriptions. These descriptions are able to provide more fine-grained information for the CLIP model to discern. The model also requires an object detection module to understand the objects inside the image.

The experimental results show that the proposed method can achieve better scores for OOD detection for multi-modal aligning. However, as reviewer qf8F pointed out: Utilizing a third-party model to create descriptors seems to dilute the advantages of the model. The proposed method is indeed a pipeline, consisting of several modules together. This can greatly increase the complexity of the problem. Such a drawback makes the method a bit limited in terms of novelty.

Based on the current limitation, we would recommend maybe accepting to Findings.

---

### Decision · Program_Chairs · 2023-10-07

**Decision:**

Accept-Findings

**Comment:**

This paper is a study on a multimodal out-of-distribution detection task, where it addresses the problem using a combination of text and image modalities with the help of a vision language model. With the help of large language models, we are able to imagine some of the image descriptions. These descriptions are able to provide more fine-grained information for the CLIP model to discern. The model also requires an object detection module to understand the objects inside the image.

The experimental results show that the proposed method can achieve better scores for OOD detection for multi-modal aligning. However, as reviewer qf8F pointed out: Utilizing a third-party model to create descriptors seems to dilute the advantages of the model. The proposed method is indeed a pipeline, consisting of several modules together. This can greatly increase the complexity of the problem. Such a drawback makes the method a bit limited in terms of novelty.

Based on the current limitation, we would recommend maybe accepting to Findings.